# Nutritional Quality of Pre-Packaged Foods in China under Various Nutrient Profile Models

**DOI:** 10.3390/nu14132700

**Published:** 2022-06-29

**Authors:** Yuan Li, Huijun Wang, Puhong Zhang, Barry M. Popkin, Daisy H. Coyle, Jingmin Ding, Le Dong, Jiguo Zhang, Wenwen Du, Simone Pettigrew

**Affiliations:** 1The George Institute for Global Health, Beijing 100600, China; yli@georgeinstitute.org.cn (Y.L.); jding@georgeinstitute.org.cn (J.D.); ldong@georgeinstitute.org.cn (L.D.); 2The George Institute for Global Health, Faculty of Medicine, University of New South Wales, Sydney, NSW 2052, Australia; dcoyle@georgeinstitute.org.au (D.H.C.); spettigrew@georgeinstitute.org.au (S.P.); 3National Institute for Nutrition and Health, Chinese Centre for Disease Control and Prevention, Beijing 100050, China; wanghj@ninh.chinacdc.cn (H.W.); zhangjg@ninh.chinacdc.cn (J.Z.); duww@ninh.chinacdc.cn (W.D.); 4Department of Nutrition, Gillings School of Global Public Health, and Carolina Population Center, University of North Carolina at Chapel Hill, Chapel Hill, NC 27516, USA; popkin@unc.edu

**Keywords:** nutrient profiling, labelling, front-of-package labels, warning labels, processed foods, ultra-processed foods, pre-packaged foods, food policy

## Abstract

This study used various nutrient profile models (NPMs) to evaluate the nutritional quality of pre-packaged foods in China to inform future food policy development. Nutrition data for pre-packaged foods were collected through FoodSwitch China in 2017–2020. The analyses included 73,885 pre-packaged foods, including 8236 beverages and 65,649 foods. Processed foods (PFs) and ultra-processed foods (UPFs) accounted for 8222 (11.4%) and 47,003 (63.6%) of all products, respectively. Among the 55,425 PFs and UPFs, the overall proportion of products with an excessive quantity of at least one negative nutrient was 86.0% according to the Chilean NPM (2019), 83.3% for the Pan American Health Organization NPM (PAHO NPM), and 90.6% for the Western Pacific Region NPM for protecting children from food marketing (WPHO NPM), respectively. In all NPMs, 70.4% of PFs and UPFs were identified as containing an excessive quantity of at least one negative nutrient, with higher proportions of UPFs compared to PFs. Food groups exceeding nutrient thresholds in most NPMs included snack foods, meat and meat products, bread and bakery products, non-alcoholic beverages, confectionery, and convenience foods. In conclusion, PFs and UPFs accounted for three-fourths of pre-packaged foods in China, and the majority of PFs and UPFs exceeded the threshold for at least one negative nutrient under all three NPMs. Given the need to prevent obesity and other diet-related chronic diseases, efforts are warranted to improve the healthiness of foods in China through evidence-based food policy.

## 1. Introduction

Globally, there has been a rapid increase in rates of overweight and obesity in children, adolescents, and adults over the past four decades, which has caused a tremendous rise in disease burden [1,2]. These changes have been driven by nutrition transitions characterized by shifts in diet due to the modernization of food systems and associated rise in the availability and promotion of packaged and processed foods that are often high in energy, saturated fat, added sugars, and sodium [3]. Ultra-processed foods (UPFs), which are manufactured products formulated from food-derived substances along with additives and other industrially produced ingredients, are also a concern [4]. Sales of UPFs are growing in all regions around the world, but most rapidly in upper-middle and lower-middle income countries [5]. Mounting evidence shows that the consumption of UPFs is associated with obesity, cardiovascular diseases, some types of cancer, and depression [6,7]. 

Nutrient profiling systems have been developed as a method for assessing the nutritional quality of food and beverage products according to their energy content and nutrient composition, and underpin a wide range of food policies including food reformulation, food labelling, and food marketing [8,9,10]. The World Health Organization (WHO) has published a series of nutrient profile models for different regions including a nutrient profile model for the Western Pacific Region to protect children from food marketing (WPHO NPM) and a Pan American Health Organization Nutrient Profile Model (PAHO NPM) [11,12]. Many countries have developed NPMs to promote healthy diets such as implementing front-of-pack nutrition labelling (FOPL) to nudge consumers and industry towards healthier products [13]. A well-known example is Chile’s Law of Food Labelling and Advertising, enacted in 2016, requiring a warning FOPL for products high in sodium, total sugars, saturated fats, and/or total energy (the Chilean NPM) [14,15]. The introduction of this law was associated with a 24% drop in sugary drink purchases and significant reductions in the proportion of “high in” products across several food groups [16,17]. 

Studies of many countries have used existing NPMs to evaluate the nutritional quality of pre-packaged foods in local food supplies and to compare the NPMs with the intention to examine the adaptability of the NPMs. The PAHO NPM and Chilean NPM were widely used in studies of Latin American countries such as Columbia, Brazil, Mexico, and Honduras to inform the development of their own NPM [18,19,20,21]. The two NPMs were also adopted in a recent study in South Africa [22]. The Chilean NPM was used by a study in India for comparison with the South and East Asia NPM (SEARO) [23]. The PAHO NPM was also used by a study in Canada to compare the NPMs aiming to restrict the commercial marketing of foods and beverages to children [24]. Although these studies varied in the number of food products from over 1000 to more than 10,000 and in the prevalence of foods containing an excessive quantity of nutrients, from 40% to 100%, all contributed to the understanding of the healthiness of the global food supply. 

China has a high prevalence of overweight and obesity: 50.7% of adults aged 18–69 years, 19% of adolescents aged 6–17 years, and 10.4% of children under 6 years were classified as overweight or obese in 2018 [25,26]. Given the size of the population, the prevalence of overweight and obesity in China was the highest worldwide. Like many other countries, the increasing prevalence of processed foods high in saturated and trans fat, sodium, and added sugars is a major driver of this obesity epidemic [27,28,29]. To help tackle these unhealthy food environments, effective and evidence-based policy actions are needed [30]. In 2013, China enacted the national standard for nutrition labelling of pre-packaged foods, which constitutes the foundation of the NPM. The regulation (i) requires the mandatory labelling of energy, protein, total fat, carbohydrates, and sodium in the nutrition information panels on food packaging and (ii) states the nutrient reference values for these nutrients (China NRV) and requires that the amount as a percentage of the NRV is calculated and listed on the nutrition label (NRV%) [31]. In 2018, the Chinese Nutrition Society developed a voluntary FOP in the form of a Healthy Choice logo for 10 food groups [32]. However, it was poorly implemented due to lack of government support. 

As the country with the largest population in the world, China has a huge food production and consumption market. The healthiness of food supply is connected with the global health and development. China was ranked in the least 3rd among 12 countries for healthy beverages and foods according to the Health Star Rating NPM for pre-packaged foods with the data collected in 2013–2018 [30]. Our previous studies have reported high levels of sodium in several food groups, such as sauces and processed meat and fish products [33,34]. However, the overall healthiness of the pre-packaged food supply remains unknown. The aim of the present study was to apply the widely used PAHO NPM, Chilean NPM, and WPHO NPM (which covers the China region) to evaluate the nutritional quality of pre-packaged foods in China to inform future national food policy development. This study will further enrich body of NPM literature worldwide and add a lens to the understanding of the healthiness of the global food supply. 

## 2. Materials and Methods

### 2.1. Data Collection

The FoodSwitch database contains nutrient and ingredient list information for pre-packaged foods available for sale in food market [35,36]. This cross-sectional study used 2017–2020 data from the FoodSwitch China database. 

In 2017–2018, data collection took place in two provincial capital cities (Shijiazhuang and Chengdu) located in northern and southern areas of China, respectively. Students from Hebei Medical University and Sichuan University were trained to collect data from major supermarkets in the cities by using a bespoke smart-phone application. First, the bar code of a pre-packaged food was scanned to link the product. Then, photographs of the food were taken to capture all the information on the package [36]. Since 2019, data have been crowdsourced from consumers nationwide who upload the bar code and food images through FoodSwitch China WeChat Applet. The information relating to nutrients and ingredients that is captured in the images is entered into a central data management system and double-checked by trained staff. Ethics committee approval was not required for this study. 

### 2.2. Data Categorization

Products were classified according to a standard food categorization system developed by The Global Food Monitoring Group that classifies products into 18 major food groups [37]. Four food groups were excluded from analyses: vitamins and supplements, alcohol, special foods, and foods unable to be categorized. These food groups were excluded as they are either not commonly consumed or do not require food labelling in China. The 14 eligible food groups used in this study included: bread and bakery products, cereal and cereal products, confectionery, convenience food, dairy products, edible oil and oil emulsions, egg and egg products, fish and fish products, fruits and vegetables, meat and meat products, non-alcoholic beverages, sauces and spreads, snack foods, and sugars and honeys. The description of the 14 food groups is shown in Appendix A.

In addition, foods were classified according to their level of processing as defined by the NOVA food classification system, which categorizes foods into four groups: (1) unprocessed or minimally processed foods; (2) processed culinary ingredients; (3) processed foods (PFs); and (4) ultra-processed foods (UPFs) [38]. In this study, foods of plant or animal origin without added salts, sugars, fats, sweeteners, and additives were classified as unprocessed or minimally processed foods; foods used as cooking materials or condiments were classified as processed culinary ingredients, such as oils, sugars, starches, salts, and sauces; foods with added salts, sugars, or fats but not sweeteners and additives in ingredients were classified as PFs; and foods with sweeteners or additives were classified as UPFs [39]. The definition of added sugar, salt, fat, non-sugar sweetener (NSS), and food additives beyond NSS is based on whether the ingredients contain relevant keywords, which was in reference to literatures and adapted by researchers according to the actual ingredients of pre-packaged foods in China (as shown in Appendix A) [19,40].

### 2.3. Inclusion and Exclusion Criteria

All pre-packaged foods and beverages with complete nutrition information panels (NIPs) and ingredients were included. According to the regulation in China, energy, protein, total fat, carbohydrates, and sodium must be listed in NIPs, and trans fat should be listed if present in the food [31]. 

In addition to the exclusion of the four food groups mentioned previously, foods with incorrect or incomplete nutrition information, duplicate products (same product in the same package size), and foods that were unable to be categorized or were missing ingredient information were excluded from the analysis. 

### 2.4. Criteria Used for Nutrient Assessment

We chose the PAHO NPM, Chilean NPM (2019), WPHO NPM, and China NRV as the criteria for assessment. Appendix A show the details of these criteria. 

#### 2.4.1. China NRV

The China NRV is used in food nutrition labels as a reference to describe the nutrient content of pre-packaged foods in the form of the proportion of NRV per 100 g or per serving (NRV%) [31]. The proportion of per 100 g was uniformly used in this study. Based on the dietary reference intakes of Chinese residents, it provides daily reference intake values for most nutrients of concern, including total fat, saturated fat, trans fat, sodium, and energy. We also assessed the reference value of sugar from the Chinese dietary guidelines, which recommends less than 50 g/day of added sugars [41]. Given that the World Health Organization focus on free sugar instead of added sugar, this study used free sugars instead of added sugars for assessment [42].

#### 2.4.2. Chilean NPM

The Chilean NPM enacted staggered criteria for the year of 2016, 2018, and 2019. This study used nutrition data from the latest model released in 2019 [14,15]. For foods with added fat, sugar, or sodium, it regulates that: in the case of solid foods (those labelled in grams), the percentage of their weight is ≥0.4% for sodium, ≥10% for sugars, ≥4% for saturated fats, and their energy density is ≥275 kcal/100 g; in the case of liquid foods (those labelled in millilitres), the percentage of their weight is ≥0.1% for sodium, ≥5% for sugars, ≥3% for saturated fats, and their energy density is ≥75 kcal/100 g. As most PFs and UPFs are eligible according to the criteria, this study applied the Chilean NPM to all PFs and UPFs.

#### 2.4.3. PAHO NPM

The PAHO NPM includes total fat, saturated fat, free sugar, and sodium. For free sugar, we estimated the values from total sugars based on the method recommended by PAHO [11]. In addition to these critical nutrients, “other sweeteners”, i.e., NSS, were included in the model. As shown in Appendix A, foods reporting any 1 of the 21 sweeteners in the list of ingredients using either their scientific or trade names were regarded as containing NSS. The eligible foods as rated by the PAHO NPM include PFs and UPFs and exclude unprocessed foods and food condiments. Therefore, this study applied PAHO NPM to all PFs and UPFs.

#### 2.4.4. WPHO NPM

The WPHO NPM was developed as a tool to protect children from food marketing [12]. It consists of 18 food categories covering seven critical components of food including energy, total fat, saturated fat, total sugar, added sugar, NSS, and sodium. Three food groups (chocolate and sugar confectionery, energy bars, and sweet toppings and desserts; cakes, sweet biscuits and pastries, other sweet bakery products, dry mixes for making such; energy drinks, tea, and coffee) are not permitted to be marketed according to the NPM. Other food categories have thresholds for at least two of these components, and marketing is prohibited if the food exceeds any of these thresholds [43]. This study applied WPHO NPM to PFs and UPFs to be in line with the other NPMs. 

### 2.5. Statistical Analyses

Nutrient content per serving was uniformly calculated as per 100 g/100 mL. Foods with multiple NIPs, such as cookie sets with various flavours or convenience noodles with separate nutrition information for noodles and condiments, were calculated by taking the weighted nutrient contents according to the stated serving size on the NIP of each food product. For reconstituted foods such as milk powder, instant drinks, or soups, we used nutrition information for the product “as sold” because nutrition information “as consumed” was not available for most foods. 

The numbers and proportions of foods with negative nutrients were described by food category. The nutrient content per 100 g/100 mL and the NRV% of each nutrient were described using median and inter-quantile range (IQR). Nutrient contents of saturated fat and total sugar were based on non-missing values. The respective cutoffs for the three NPMs were used to identify the numbers and proportions of foods and beverages that exceeded criteria overall and in each food category. Foods were regarded as containing excess negative nutrients if any of the assessed nutrients exceeded the threshold specified by the NPM. This was calculated by using the number of all food products with the nutrient information as the denominator. The results for the three NPMs were compared for each food category and nutrient. 

Chi-square tests were performed to test differences in proportions of products with excessive quantities of negative nutrients between beverages and foods, and PFs and UPFs. A two-sided *p* value of <0.05 was considered statistically significant. Agreement percentages for each pair of NPMs were compared with McNemar’s tests and presented with the percentage of consistent results and kappa value. The higher the Kappa value corresponds to the higher degree of agreement. All analyses were conducted using SAS Enterprise Guide 8.3 (SAS Institute Inc., Cary, NC, USA).

## 3. Results

### 3.1. Characteristics of Ingredients and Nutrients of Pre-Packaged Foods

Of the 80,106 pre-packaged foods for which data were collected in China between 2017 and 2020, 6221 were excluded due to data error or incomplete nutrition information (*n* = 502), duplicates (*n* = 2379), missing values (*n* = 1461), and representing an ineligible food group (*n* = 1879). Finally, 73,885 products were included comprising 8236 beverages and 65,649 foods. A total of 55,425 PFs and UPFs were further included for the analysis of the three NPMs (Figure 1). 

Of the 73,885 assessed products, those with sugar, sodium, and fat in the ingredients accounted for 73.7%, 57.5%, and 48.2%, respectively. The total number of foods containing either sugar, sodium, or fat ingredients accounted for 86.5%. In addition, 21.0% of products were identified as containing NSSs, and food additives other than sweeteners were found in 71.9% of products. The top five food groups with sugar/sodium/fat ingredients were bread and bakery products (99.6%), snack foods (98.5%), meat and meat products (98.4%), convenience food (97.4%), and confectionery (96.7%). For beverages, 85.9% contained sugar ingredients and 38.3% contained NSSs. For solid foods, 72.2% contained sugars, 62.4% contained sodium, and 51.9% contained fat. According to the NOVA group system, 63.6% of products were ultra-processed, and 11.4% were processed. Unprocessed or minimally processed products accounted for 10.5%, and the remaining 14.5% were categorized as processed culinary ingredients. The proportion of products containing negative nutrients within each NOVA group was in line with the definition, with almost all UPFs containing a food additive (99.9%) and sugar/sodium/fat (98.7%), and 30.0% containing NSSs (Table 1).

Table 2 and Table 3 display the content and NRV% of each nutrient by food category. In medians, per 100 g pre-packaged foods contained 1467 kJ/100 g of energy, accounting for 17.5% of the daily energy reference intake value. Six food groups had a median sodium content higher than 600 mg/100 g, including sauces and spreads, meat and meat products, convenience foods, egg and egg products, fish and fish products, and snack foods. Of the 73,885 food products, 70,506 were missing saturated fat and 63,057 missing total sugar. Among the 3379 foods reporting the content of saturated fat, the median content of saturated fat was 5.1 g/100 g, and the median NRV% was 102.8%. Confectionery, bread and bakery foods, convenience foods, and meat products were ranked high in containing saturated fat. For the 10,828 foods with values of total sugar, the median content was 11.2 g/100 g and the median NRV% was 22.0%. Food groups high in total sugar were bread and bakery products, confectionery, non-alcoholic beverages, fruits and vegetables, and meat and meat products. Foods showed higher contents than beverages numerically in all negative nutrients. PFs and UPFs showed higher contents than unprocessed or minimally processed foods numerically in fat, saturated fat, and sodium. 

### 3.2. Nutritional Quality of Pre-Packaged Foods under Different NPMs

Table 4 shows the proportion of PFs and UPFs with negative nutrients in excess under different NPMs. Of the three NPMs, the WPHO NPM had the highest proportion of products with negative nutrients in excess (90.6%), followed by the Chilean NPM (86.0%) and the PAHO NPM (83.3%). Approximately 98.4% of products were identified as having excessive quantities of negative nutrients in at least one of the three NPMs, and 70.4% of foods were found to have excessive negative nutrients in all three models. Beverages showed higher excessive rates than foods in the PAHO and WPHO NPMs (both *p* < 0.0001), but lower rates than foods in the Chilean NPM (*p* < 0.0001). UPFs also showed higher rates than PFs in the PAHO and WPHO NPMs (both *p* < 0.0001), but the differences were nonsignificant in the Chilean NPM (*p* = 0.116). Each NPM found over 80% prevalence of products containing at least one negative nutrient in excess for 8 out of 14 food groups. The top five food groups with an excessive quantity of negative nutrients in all NPMs were snack foods (93.6%), meat and meat products (83.8%), bread and bakery products (80.4%), non-alcoholic beverages (78.5%), and convenience food (75.6%). As shown in Table 5, the agreement percentages of these five food groups were all over 80% across all pairs of NPMs. The total agreement percentages between each pair of NPMs were 85.0% for the Chilean vs. WPHO (kappa = 0.28), 77.3% for the Chilean vs. PAHO (kappa = 0.13), and 81.5% for the PAHO vs. WPHO (kappa = 0.20). The proportions of products with excessive quantities of each nutrient for each food category under the Chilean, PAHO, and WPHO NPMs are shown in Appendix A, respectively. 

## 4. Discussion

Using China NRV and three different NPMs to evaluate the nutritional quality of over 70,000 pre-packaged foods in China, this study found that most food products contained at least one negative nutrient in excessive quantities. Over 95% of products contained added sugar, sodium, or fat. The food products per 100 g/mL in medians provided approximately 17% of daily energy, 12% of daily fat, 10% of daily sodium, 103% of daily saturated fat, and 22% of total sugar. Of all the food products, 75% were PFs (11.4%) and UPFs (63.6%). Across the 55,425 PFs and UPFs assessed by the three NPMs, under the Chilean NPM 2019, 86% of foods and beverages would be required to display a black and white octagonal “high in” warning label. Moreover, 83.3% of PFs and UPFs would be regarded as high in any nutrient according to the PAHO NPM, and 90.6% would be prohibited from being marketed to children under the WPHO NPM. 

The high rates of excess negative nutrients across these diverse NPMs are likely related to the high proportion of UPFs. Studies have shown that UPFs are associated with overweight and obesity, type 2 diabetes, depression, cardiovascular and cerebrovascular disease, and mortality [6,7,44]. The negative impact on health may be associated with the poor nutritional quality of UPFs. As testified in this study, most UPFs are recognized as high in energy, sodium, saturated fat, sugar, or containing NSS. UPFs provide half or more of total energy intake in developed countries [45]. In China, the proportion of energy from processed foods increased rapidly from 9% to 30% between 1990 and 2019, and the mean intake in UPF consumption among Chinese adults increased from 12.0 g in 1997 to 41.5 g in 2011, which was in parallel with rapid increase in the prevalence of obesity [27,28]. It is noteworthy that sales of UPFs are increasing in China [3]. Improving the healthiness of pre-packaged foods in China is a public health priority that requires urgent action. 

This study identified that proportions of products with excessive quantities of negative nutrients varied among the NPMs. Overall, the WPHO NPM was the strictest, followed by the Chilean and PAHO NPMs. In most previous studies, the PAHO NPM has resulted in identification of a larger proportion of foods containing excessive numbers of nutrients than the Chilean NPM [18,19,20,22,23]. For instance, a study in Mexico showed the non-compliance rates were 97.7% under the PAHO NPM and 89.1% under the Chilean NPM 2019 [20], and a recent study in South African reported 73.2% and 64.4% for the PAHO and Chilean NPMs, respectively [22]. The higher excessive rate of the Chilean NPM than the PAHO NPM among PFs and UPFs in this study is likely related to the high excessiveness of energy content identified by the Chilean NPM. More than 80% of PFs and UPFs in the Chinese food supply contained at least one nutrient in excess under the PAHO and Chilean NPMs; this is one of the highest rates globally [18,19,20,22,23]. This result echoes that of a previous study that found China to have among the lowest levels of healthy food across 12 countries, indicating the need to improve the food environment in the country [30]. 

Our analyses also found that some food groups were consistently listed in the top 5 foods exceeding nutrient thresholds, including snack foods, meat and meat products, bread and bakery products, non-alcoholic beverages, confectionery, and convenience foods. These results are consistent with those of other studies [19,21,46], which is concerning given the increasing consumption of these foods in China [47]. Moreover, although the assessment of the three NPMs did not evaluate the nutritional quality of processed culinary ingredients (i.e., sauces), the high content of sodium in culinary ingredients is of concern. Previous research indicates that sauces account for about 10% of salt intake in China [48], which is likely to be partly due to the high sodium content of sauces in China, as the sauces in China contained more than four times as much sodium compared to equivalents in the UK [33]. This highlights the importance of including culinary ingredients in NPMs in the Chinese context. 

In addition, the PAHO and WPHO NPMs include NSSs as a nutrient to limit. In the current study, NSSs were found in one-fifth of the assessed 73,885 products, which is less than the 55% reported in a study in Chile [49], but much higher than the value found in studies conducted in Colombia, Brazil, Australia, and Hong Kong, where the proportion of products containing NSSs was no higher than 16% [18,19,50,51]. NSSs are usually added to products as (in part) a replacement for added sugars in response to emerging sugar reduction policies and increasing consumer concern regarding added sugars [52], but their impact on health remains controversial [53]. It was reported by WHO that replacing sugars with NSSs in the short term results in reductions in body weight, but may be associated with increased risk of type 2 diabetes, cardiovascular diseases, and mortality in the longer term [54]. The wide usage of NSSs found in this study deserves further research attention.

Currently, pre-packaged food contributes ~30% of total energy intakes and 13.5% of sodium intakes in China [27,48]. Actions to facilitate a healthy food supply and encourage healthy food choices are lagging behind the fast-growing consumption of nutrient-poor, energy-dense pre-packaged foods. The findings of this study have important policy implications. In line with other countries [9,55], there is much more the government can do to create a heathier pre-packaged food supply in China. This includes implementation of interpretive front-of-pack labels on food packages that effectively convey information about foods’ nutritional quality [56], setting national nutrient reformulation targets for food manufacturers, and/or implementation of taxes for unhealthy foods. In addition, the public should be advised and educated on how to read nutrition informational panels and how to choose minimally processed foods and avoid the consumption of UPFs as much as possible. 

This study had a few strengths. First, we used a large sample of pre-packaged foods available for sale in Chinese supermarkets over the last four years. It was demonstrated that small numbers of food products would increase the bias of the results [23]. The over 70,000 food products collected from a single country appear to represent the largest number used for nutritional assessment under NPMs. Second, we made full use of the nutritional information panel and the ingredient information, and undertook a comprehensive analysis of the nutritional quality of these products by assessing the nutrient content and the overall healthiness according to a range of global NPMs. In addition, we adopted NOVA groups in food categories and compared the nutritional quality of PFs and UPFs. As such, these findings provide important insights into how PFs and UPFs score across different NPMs.

This study had some limitations to mention. First, there were a considerable number of missing data for saturated fat and total sugar because these nutrients are not required to be declared in the NIP in China. Therefore, the excessive rates of saturated fat and total sugar were based on a relatively small number of products. The total excessive rate might be underestimated as those foods potentially high in the missing nutrients were not included in the numerators. Second, for a small proportion of reconstituted foods (1893 food products, accounting for 2.6% of the sample), we used the nutrition information “as sold” rather than “as consumed” because the latter was not available in nutrition labels, which might have resulted in some bias in estimating the consumed nutrient content. Lastly, the three NPMs were not developed specifically for China, and their potential adaptability for use in China is yet to be verified. However, the current study demonstrates that these NPMs can be applied to pre-packaged foods in China, and the findings provide valuable information about the healthiness of the food supply in China and highlight the need for effective strategies to reduce high levels of negative nutrients and the high prevalence of UPFs. 

## 5. Conclusions

PFs and UPFs accounted for three-fourths of pre-packaged foods in China. The majority of PFs and UPFs exceeded the threshold for at least one negative nutrient under three different NPMs, especially in the food groups of snack foods, meat and meat products, bread and bakery products, non-alcoholic beverages, confectionery, and convenience foods. The high prevalence of unhealthy pre-packaged foods in China highlights the need for stronger and evidence-based policies to improve the food environment to help prevent the high prevalence of obesity and other diet-related chronic diseases.

## Figures and Tables

**Figure 1 nutrients-14-02700-f001:**
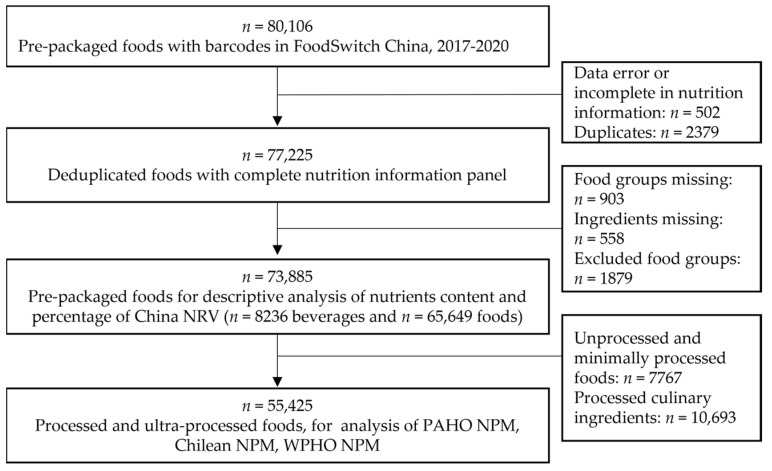
Flowchart of the data for analysis. NRV, nutrient reference value; NPM, nutrient profile model; WPHO, World Health Organization Western Pacific; PAHO, Pan American Health Organization.

**Table 1 nutrients-14-02700-t001:** Proportion of food products with sugar, sodium, fat, and food additives in ingredients.

Category	Number of Products	Sugar *n* (%)	Sodium *n* (%)	Fat *n* (%)	NSS *n* (%)	Additives *n* (%)	Sugar/Sodium/Fat *n* (%)
Food groups							
Bread and bakery products	10,380	10,025 (96.6)	8871 (85.5)	9766 (94.1)	2253 (21.7)	9755 (94.0)	10,338 (99.6)
Cereal and cereal products	6341	1521 (24.0)	2296 (36.2)	1121 (17.7)	316 (5.0)	1845 (29.1)	2921 (46.1)
Confectionery	5328	5077 (95.3)	1107 (20.8)	2823 (53.0)	1387 (26.0)	5122 (96.1)	5151 (96.7)
Convenience food	3198	2781 (87.0)	2797 (87.5)	2665 (83.3)	593 (18.5)	2787 (87.1)	3115 (97.4)
Dairy products	5592	4481 (80.1)	1256 (22.5)	2286 (40.9)	1721 (30.8)	4516 (80.8)	4783 (85.5)
Edible oil and oil emulsions	1119	4 (0.4)	42 (3.8)	995 (88.9)	1 (0.1)	81 (7.2)	997 (89.1)
Egg and egg products	353	140 (39.7)	277 (78.5)	37 (10.5)	2 (0.6)	163 (46.2)	278 (78.8)
Fish and fish products	1222	733 (60.0)	1042 (85.3)	675 (55.2)	48 (3.9)	682 (55.8)	1049 (85.8)
Fruits and vegetables	13,262	8241 (62.1)	6447 (48.6)	3222 (24.3)	3577 (27.0)	7864 (59.3)	10,141 (76.5)
Meat and meat products	5185	4571 (88.2)	5085 (98.1)	2209 (42.6)	160 (3.1)	4632 (89.3)	5104 (98.4)
Non-alcoholic beverages	6194	5296 (85.5)	1071 (17.3)	1030 (16.6)	1965 (31.7)	4953 (80.0)	5393 (87.1)
Sauces and spreads	7525	4190 (55.7)	6200 (82.4)	3399 (45.2)	1390 (18.5)	5433 (72.2)	6617 (87.9)
Snack foods	6319	5612 (88.8)	5953 (94.2)	5405 (85.5)	2058 (32.6)	5185 (82.1)	6222 (98.5)
Sugars and honeys	1867	1774 (95.0)	20 (1.1)	8 (0.4)	38 (2.0)	83 (4.4)	1775 (95.1)
Foods or Beverages							
Beverages	8236	7073 (85.9)	1515 (18.4)	1582 (19.2)	3151 (38.3)	6810 (82.7)	7199 (87.4)
Foods	65,649	47,373 (72.2)	40,949 (62.4)	34,059 (51.9)	12,358 (18.8)	46,291 (70.5)	56,685 (86.3)
NOVA groups							
Unprocessed or minimally processed foods	7767	0 (0)	0 (0)	0 (0)	0 (0)	155 (2.0)	0 (0)
Processed culinary ingredients	10,693	5981 (55.9)	6277 (58.7)	4406 (41.2)	1430 (13.4)	5626 (52.6)	9411 (88.0)
Processed foods	8422	5276 (62.6)	5013 (59.5)	3400 (40.4)	0(0)	348 (4.1)	8078 (95.9)
Ultra-processed foods	47,003	43,189 (91.9)	31,174 (66.3)	27,835 (59.2)	14,079 (30.0)	46,972 (99.9)	46,395 (98.7)
Total	73,885	54,446 (73.7)	42,464 (57.5)	35,641 (48.2)	15,509 (21.0)	53,101 (71.9)	63,884 (86.5)

NSS = non-sugar sweetener.

**Table 2 nutrients-14-02700-t002:** Nutrient content of food products by food categories, median (IQR).

Category	N1 *	Energy (kJ/100 g)	Fat (g/100 g)	Sodium (mg/100 g)	N2 *	Saturated Fat (g/100 g)	N3 *	Total Sugar (g/100 g)	Free Sugar (g/100 g)
Food groups									
Bread and bakery products	10,380	1979 (444)	20.9 (12.3)	223 (215)	640	9.5 (8.9)	617	15.1 (23.3)	15.1 (23.3)
Cereal and cereal products	6341	1496 (118)	1.5 (4.4)	20 (270)	313	0.5 (1.5)	317	3.5 (14.6)	3.5 (10.5)
Confectionery	5328	1667 (593)	1.3 (22)	45 (90)	332	15.4 (15.6)	540	0.0 (29)	0.0 (29)
Convenience food	3198	1296 (965)	10.1 (14.6)	848 (1544)	84	6.9 (9.2)	63	2.8 (1.5)	2.8 (1.5)
Dairy products	5592	382 (749)	3.5 (8.5)	60 (52)	229	5.1 (9)	293	5.0 (12.5)	2.5 (6.2)
Edible oil and oil emulsions	1119	3700 (14)	99.9 (0.5)	0 (0)	481	13.0 (5)	10	0.0 (0.6)	0.0 (0.6)
Egg and egg products	353	677 (183)	10.5 (3.5)	714 (737)	0		0		
Fish and fish products	1222	737 (788)	6.7 (14.4)	701 (758)	34	2.5 (2.2)	36	2.4 (7)	2.4 (7)
Fruits and vegetables	13,262	1384 (1155)	2.0 (27.5)	131 (697)	204	0.6 (2.1)	234	8.1 (37.2)	5.2 (18.3)
Meat and meat products	5185	963 (736)	9.6 (8.7)	1086 (713)	41	6.3 (6.2)	41	5.9 (21.7)	5.9 (21.7)
Non-alcoholic beverages	6194	190 (1004)	0.0 (0.9)	21 (43)	336	0.0 (0)	6190	10.3 (41)	10.3 (41)
Sauces and spreads	7525	733 (1234)	3.5 (23.2)	2420 (5115)	257	0.6 (8.2)	183	2.6 (8.4)	2.6 (8.4)
Snack foods	6319	2005 (588)	22.5 (17.1)	658 (768)	395	5.0 (5)	439	2.1 (2.1)	2.1 (2.1)
Sugars and honeys	1867	1419 (300)	0.0 (0.6)	18 (30)	33	0.0 (0)	1865	81.6 (19)	81.6 (19)
Foods or Beverages									
Beverages	8236	200 (175.3)	0.0 (2.5)	30 (47)	447	0.0 (1)	5037	8.9 (6.7)	8.7 (6.6)
Foods	65,649	1516 (989)	9.0 (22.2)	260 (859)	2932	7.0 (11.5)	5791	54.1 (74.3)	53.0 (74.6)
NOVA groups									
Unprocessed or minimally processed	7767	1460 (323)	1.6 (3.9)	10 (50)	215	0.5 (1.6)	727	15.8 (58.9)	15.8 (59.4)
Processed culinary ingredients	10,693	1300 (1298.7)	2.3 (33.3)	800 (4154)	772	10.3 (12.3)	2059	80.0 (20.6)	80.0 (20.6)
Processed foods	8422	1516 (1033)	7.4 (22.9)	184 (576)	346	2.0 (6.6)	1012	11.6 (38.4)	11.5 (35.8)
Ultra-processed foods	47,003	1493 (1269)	9.8 (21.1)	235 (692)	2046	5.1 (11)	7030	9.0 (16.6)	8.6 (15.5)
Total	73,885	1467 (1218)	7.0 (20.9)	192 (743)	3379	5.1 (11.7)	10,828	11.2 (62.8)	11.0 (63)

* N1: number of products for energy, fat, and sodium; N2: number of products for saturated fat; N3: number of products for total sugar and free sugar.

**Table 3 nutrients-14-02700-t003:** Percentage of the nutrient content with China NRV by food categories, median (IQR).

Category	N1 *	Energy	Fat	Sodium	N2 *	Saturated Fat	N3 *	Free Sugar
Food groups								
Bread and bakery products	10,380	23.6 (5.3)	34.8 (20.5)	11.2 (10.8)	640	178.2 (144.1)	617	30.2 (46.6)
Cereal and cereal products	6341	17.8 (1.4)	2.5 (7.3)	1.0 (13.5)	313	14.3 (33.4)	317	7.0 (21.2)
Confectionary	5328	19.9 (7.1)	2.2 (36.7)	2.3 (4.5)	332	258.0 (230.1)	540	0.0 (58)
Convenience food	3198	15.4 (11.5)	16.8 (24.3)	42.5 (77.2)	84	151.4 (113.9)	63	5.6 (3)
Dairy products	5592	4.5 (8.9)	5.8 (14.2)	3.0 (2.6)	229	258.7 (201.2)	293	5.0 (12.5)
Edible oil and oil emulsions	1119	44.0 (0.1)	166.5 (0.9)	0.0 (0)	481	130.4 (50)	10	0.0 (1.2)
Egg and egg products	353	8.1 (2.1)	17.5 (5.9)	35.7 (36.8)	0		0	
Fish and fish products	1222	8.8 (9.3)	11.2 (24)	35.1 (37.9)	34	118.3 (72.8)	36	4.8 (13.9)
Fruits and vegetables	13,262	16.5 (13.8)	3.3 (45.8)	6.6 (34.8)	204	59.2 (94.1)	234	10.3 (36.7)
Meat and meat products	5185	11.5 (8.7)	16.0 (14.5)	54.3 (35.6)	41	135.6 (198.8)	41	11.8 (43.5)
Non-alcoholic beverages	6194	2.3 (12)	0.0 (1.5)	1.1 (2.1)	336	0.0 (0)	6190	20.6 (82)
Sauces and spreads	7525	8.7 (14.7)	5.8 (38.7)	121.0 (255.7)	257	37.7 (128.6)	183	5.2 (16.8)
Snack foods	6319	23.9 (7)	37.5 (28.5)	32.9 (38.4)	395	83.3 (114.6)	439	4.2 (4.2)
Sugars and honeys	1867	16.9 (3.6)	0.0 (1)	0.9 (1.5)	33	0.0 (0)	1865	163.2 (38)
Foods or Beverages								
Beverages	8236	2.4 (2.1)	0.0 (4.2)	1.5 (2.3)	447	0.0 (112.1)	5037	17.4 (13.2)
Foods	65,649	18.0 (11.8)	15.0 (37)	13.0 (42.9)	2932	118.3 (162.1)	5791	106.0 (149.1)
NOVA groups								
Unprocessed or minimally processed	7767	17.4 (3.8)	2.7 (6.5)	0.5 (2.5)	215	12.7 (71.2)	727	31.6 (118.8)
Processed culinary ingredients	10,693	15.5 (15.4)	3.8 (55.5)	40.0 (207.7)	772	118.0 (85.6)	2059	160.0 (41.2)
Processed foods	8422	18.0 (12.3)	12.3 (38.2)	9.2 (28.8)	346	59.2 (134.6)	1012	22.9 (71.5)
Ultra-processed foods	47,003	17.8 (15.1)	16.3 (35.1)	11.8 (34.6)	2046	125.6 (199)	7030	17.2 (31)
Total	73,885	17.5 (14.5)	11.7 (34.8)	9.6 (37.2)	3379	102.8 (173.4)	10,828	22.0 (125.9)

* N1: number of products for energy, fat, and sodium; N2: number of products for saturated fat; N3: number of products for free sugar. NRV, nutrient reference value. NRV of each nutrient: energy, 8400 kJ; fat, 60 g; saturated fat, 10% of total energy; trans fat, 2.2 g; free sugar (see Section 2.4.1 for explanation), 50 g; sodium, 2000 mg. The median and IQR of trans fat were all 0 by food categories and in total.

**Table 4 nutrients-14-02700-t004:** Proportion of food products containing at least one negative nutrient in excess under different NPMs.

Category	Number of Products	Chilean NPM *n* (%)	PAHO NPM *n* (%)	WPHO NPM *n* (%)	Any NPM *n* (%)	All NPMs *n* (%)
Food groups						
Bread and bakery products	10,355	10,057 (97.1)	8852 (85.5)	9750 (94.2)	10,324 (99.7)	8329 (80.4)
Cereal and cereal products	3077	2779 (90.3)	1856 (60.3)	1824 (59.3)	2863 (93.0)	1627 (52.9)
Confectionery	5313	4420 (83.2)	3163 (59.5)	5313 (100.0)	5313 (100.0)	2588 (48.7)
Convenience food	3128	2467 (78.9)	2790 (89.2)	2666 (85.2)	2991 (95.6)	2364 (75.6)
Dairy products	4856	2842 (58.5)	3939 (81.1)	3942 (81.2)	4529 (93.3)	2249 (46.3)
Egg and egg products	283	260 (91.9)	281 (99.3)	187 (66.1)	282 (99.6)	187 (66.1)
Fish and fish products	1049	888 (84.7)	1036 (98.8)	602 (57.4)	1040 (99.1)	597 (56.9)
Fruits and vegetables	10,319	8450 (81.9)	7578 (73.4)	9863 (95.6)	10,238 (99.2)	6457 (62.6)
Meat and meat products	5126	4885 (95.3)	5092 (99.3)	4314 (84.2)	5100 (99.5)	4297 (83.8)
Non-alcoholic beverages	5695	4506 (79.1)	5612 (98.5)	5557 (97.6)	5665 (99.5)	4471 (78.5)
Snack foods	6224	6087 (97.8)	5944 (95.5)	6207 (99.7)	6220 (99.9)	5826 (93.6)
Foods or Beverages						
Beverages	7501	5490 (73.2)	7016 (93.5)	7181 (95.7)	7453 (99.4)	5088 (67.8)
Foods	47,924	42,151 (88.0)	39,127 (81.6)	43,044 (89.8)	47,112 (98.3)	33,904 (70.7)
χ2		1171	658	267	47	26
*p* value		<0.0001	<0.0001	<0.0001	<0.0001	<0.0001
NOVA groups						
Processed foods	8422	7193 (85.4)	6220 (73.9)	6767 (80.3)	8057 (95.7)	5249 (62.3)
Ultra-processed foods	47,003	40,448 (86.1)	39,923 (84.9)	43,458 (92.5)	46,508 (98.9)	33,743 (71.8)
χ2		2	629	1232	503	307
*p* value		0.116	<0.0001	<0.0001	<0.0001	<0.0001
Total	55,425	47,641 (86.0)	46,143 (83.3)	50,225 (90.6)	54,565 (98.4)	38,992 (70.4)

**Table 5 nutrients-14-02700-t005:** Agreement between pairs of nutrient profile models by food categories.

		Chilean vs. WPHO		Chilean vs. PAHO		PAHO vs. WPHO	
Category	Number of Products	AG%	Kappa	AG%	Kappa	AG%	Kappa
Food Groups							
Bread and bakery products	10,355	93.2	0.19	84.1	0.04	84.2	0.15
Cereal and cereal products	3077	66.7	0.22	64.7	0.15	88.3	0.76
Confectionery	5313	83.2	0.00	54.7	−0.04	59.5	0.00
Convenience food	3128	93.3	0.77	83.4	0.39	83.2	0.25
Dairy products	4856	68.9	0.31	56.1	0.02	81.0	0.38
Egg and egg products	283	74.2	0.29	91.9	0.07	66.8	0.03
Fish and fish products	1049	71.8	0.37	85.1	0.08	58.6	0.03
Fruits and vegetables	10,319	83.0	0.19	71.9	0.20	71.8	0.01
Meat and meat products	5126	88.2	0.38	95.6	0.18	84.8	0.07
Non-alcoholic beverages	5695	80.5	0.12	80.4	0.10	97.2	0.27
Snack foods	6224	97.8	0.09	94.1	0.09	95.5	0.05
Foods or Beverages							
Beverages	7501	72.9	0.06	70.6	0.02	93.4	0.35
Foods	47,924	86.9	0.34	78.4	0.17	79.6	0.18
NOVA groups							
Processed foods	8422	84.2	0.45	73.8	0.21	75.3	0.31
Ultra-processed foods	47,003	85.1	0.23	77.9	0.11	82.6	0.14
Total	55,425	85.0	0.28	77.3	0.13	81.5	0.20

## Data Availability

The data that support the findings of this study are available from the corresponding author upon reasonable request and in compliance with the pertinent regulations of data management and data sharing in China.

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
