# Peer review of "Nutritional Quality of Pre-Packaged Foods in China under Various Nutrient Profile Models"

_nutrients, 2022, doi:10.3390/nu14132700_

Round 1

Reviewer 1 Report

In this manuscript, authors present a detailed analysis of ingredient and nutrient composition of pre-packaged foods. In addition, they apply different nutrient profile models in order to assess the nutritional quality of those products.

This is a potentially interesting work in line with previous studies in other countries. However, some of the methods used have to be modified in order to render a well-designed study. In addition, some major changes must be made.

General

All across the manuscript, foods exceeding nutrients according to the different NPMs are considered rated/eligible for regulation. This is incorrect since i.e. foods eligible for regulation by the PAHO NPM are those processed and UPF, as stated in the NPM document. However, foods exceeding any of the nutrients are those surpassing the thresholds stablished by the NPM. Please, modify accordingly.

Abstract

Lines 27-29: "Under all three NPMs, the majority of pre-packaged foods in China would exceed the threshold for at least one negative nutrient, especially for UPFs and PFs". This information is already provided in lines 23-25.

Introduction

·      Lines 54-55: Consider whether the following reference may be appropriate for this sentence: Labonté M, Poon T, Gladanac B, Ahmed M, Franco-Arellano B, Rayner M and L'Abbé MR, 2018. Nutrient profile models with applications in government-led nutrition policies aimed at health promotion and noncommunicable disease prevention: A systematic review. Advances in Nutrition, 9:741-788. doi:1810 10.1093/advances/nmy045

·      Line 56: include appropriate reference on the Chile's Law of Food Labelling and Advertising

Materials and Methods

This section has to be thoroughly reviewed.

·      I strongly recommend authors to include a table in supplemental material with a detailed description of each of the 14 food groups analysed. One of the limitations when comparing data from different publications is that food groups are not sufficiently characterized. Therefore, results are difficult to compare.

·      Lines 119-120 and 191-194. The presence of complete nutrition information panels in prepackaged foods is a mandatory inclusion criteria. Therefore, the percentage of foods with this information does not qualify as result and should be removed (lines 231-232, Tables 3 and S3). Please, cite the Chinese law in lines 119-120.

·      Lines 122-123: "foods with a missing food group". Do you mean foods in a missing food groups? Please, describe it in a clearer way.

·      Lines 129: "…per 100 g or per serving…". Please clarify which one was used for foods with information for both.

·      Lines 133-134: "which recommends no more than 10% of total energy from added sugars per day or less than 50 g/day added sugars". Please, clarify the criterion followed in this work and add the proper reference of the Chinese dietary guidelines.

·      Lines 134-136: "Given the high prevalence of missing values for added sugars, this study used free sugars instead of added sugars for assessment". The change in the criterion is not justified. Therefore, unless a proper justification is issued, authors should use added sugar as stated in the Chinese dietary guidelines. The justification may well be the recommendation by the World Health Organization in 2015.

·      Lines 145-147: "Natural ingredients such as table sugar, table salt, or table oil are excluded for warning labels under this NPM". How were they considered for the purpose of the Chilean NPM: as compliant with regulation? If so, please see comment regarding lines 147-148.

·      Lines 147-148: "In this study, foods and beverages without added sugar, sodium, or fat were regarded as not containing excessive quantities of the nutrients under the criteria". I strongly disagree with the authors in this criterion because it is misleading. In fact, I consider it arbitrary and unsuited for this work. Authors have two options: either they use the same criteria for all foods or they exclude foods and beverages without added sugar, sodium, or fat from the analysis. In the latter, I would suggest including them in an additional row in Table S5.

·      Lines 151-152: "For free sugar, we estimated the values from total sugars based on the method recommended by PAHO [10]". To our own experience, this method is too simple and unpractible when applied to the great diversity of foods in the market, particularly to UPF. As a consequence of this complexity, a good number of papers address the difficult task of calculating free sugars. One I find most useful is this one

o  Swan, G.E.; Powell, N.A.; Knowles, B.L.; Bush, M.T.; Levy, L.B. A definition of free sugars for the UK. Public Health Nutr. 2018, 21, 1636–1638. https://doi.org/10.1017/S136898001800085X

·      Therefore, it is hard to believe that authors didn't use any extra criteria.

·      Lines 152-153: "…the PAHO model requires that a food must display a warning if it contains “non-sugar sweeteners”…". This is not entirely correct, since warning labels is only one of the possible uses of the PAHO NPM. Instead, I suggest the sentence included in the document (page 14): "In addition to critical nutrients, “other sweeteners” were included in the model".

·      Lines 164-166: "In this study, we also assessed processed culinary ingredients, but excluded sugar for sugar regulation, salt condiments for sodium regulation, and fat and oil for total fat and saturated fat regulation". On what grounds did authors exclude these products? I see no reason for exclusion.

·      Was the PAHO NPM applied to all foods except for those mentioned in lines 164-166? Please, explain in the text.

·      Lines 170-171: "Each food category has thresholds for at least two of these components…". This sentence is incorrect, because as the authors acknowledge in lines 176-179, this is not the case for three groups. Please, modify accordingly.

·      Lines 173-174: "A small number of unprocessed foods and culinary ingredients (125 food products) that couldn’t be mapped to the model were regarded as permitted under the criteria". See comment regarding lines 147-148: I strongly recommend either to apply the same criteria as for the rest of foods or to exclude them entirely from this analysis.

·      Lines 176-178: "Three food groups (Chocolate and sugar confectionery, energy bars, and sweet toppings and desserts; Cakes, sweet biscuits and pastries, other sweet bakery products, dry mixes for making such; Energy drinks, tea and coffee) not permitted in the NPM were regarded as excessive for all the nutrient thresholds". I agree with the authors but only for data in Table 3. However, for Table 4, they cannot be included as excessive for all nutrients because they cannot be evaluated. Instead, I recommend to add a new row specific for these foods. The interpretation does not change, but data does.

·      Lines 194-196: "The analysis of each nutrient was based on the total number of products within each category that displayed this information on the nutrition information panel". This sentence should be included in Results and in the corresponding tables for clarity.

·      Lines 200-201: "This was calculated using all food products as the denominator regardless of missing values in nutrients". Was this applied to Table 4? If so, I consider it incorrect because only the sample with the nutrient information analysed should be used as the denominator.

·      Which criteria was used to determine that a product had added sodium? Were the sodium salts of additives  considered as adding sodium to the product?

·      Which criteria was used to determine that a product had added sugar and fat? Please, explain and provide a list with the ingredients listed as added sugar and fat.

·      Was any particular additives list used to determine the presence of these in foods? Please, provide.

Results

·      This section is very poorly described. Authors provide data on the text that can be easily observed in the tables. Authors present a very large amount of data that may be overlooked when improperly described. I strongly suggest to remove the duplicated values in the text. Instead, highlighting differences among food groups, NPMs, etc. should be included.

·      Lines 211: please, remove "n="

·      Lines 223-224: Shouldn't these values be the same for sugar, sodium and fat as the ones in line 217, and in line 218 for NSS? The values in lines 223-224 are not for all products, but only foods.

·      Lines 234-235: "Trans fat without reporting values could be regarded as not containing added trans-fat according to regulations in China". Does this mean that for products with trans fat it is compulsory, by law, to display the values? If so, please explain and include the reference for the Chinese law.

·      Table 1 – NOVA and lines 227-229: data provided is rather confirmation of the definitions for each of the categories, instead of a novelty. Authors should mention this concept somewhere in the text.

·      Table 2 and S4: data in this table are interesting from a national perspective, while data in Table S4 are so from an international point of view. Therefore, I strongly recommend authors to switch both tables: Table S4 in the main text and Table 2 to supplemental material.

·      Table 2 and S4: The number of foods used in the denominator for each of the values should be included, particularly in the table in the main text.

·      Table 2: Authors include mean and SD values, assuming normality in the sample distribution. Since they do not perform normality tests and means are more sensitive to extreme values, I recommend to use median and IQR values, as in Table S4.

·      Table 2 – "Note: Nutrient contents of saturated fat and added sugar was based on non-missing values". Please, use a "*" in the first raw for saturated fat and added sugar in order to improve the visualization of this important note.

·      Table 2 – Lines 241-242: "Added sugar was estimated with free sugars". This is incorrect. Since authors analyse free sugar instead of added sugar, "free sugars" must be displayed in the first row in the table, with the proper explanation as a footnote.

·      Table 3: Once a statistical significant difference was obtained for the several NOVA groups, a test should be applied to determine which groups were different (multiple comparisons tests)

·      Table 3: "Proportion of pre-packaged foods identified as containing excessive negative nutrients under different NPMs". It should say "…at least one excessive negative nutrient…"

·      Table 3 – footnote: Delete P <0.0001 because it is already in the table.

·      For the comparison of the three NPM a disagreement test is highly recommended. It would quantify the degree of similarity or diversity of the NPMs and provide an important piece of data for interpretation.

·      Table 4 – footnote: "Note: Percentage of saturated fat and total sugar was based on non-missing values". Also for trans fat. 

·      Table 4: The number of foods used in the denominator for each of the values should be included.

Discussion

·      Discussion is rather poor at interpreting own data and on relating it to previous works.

·      Is there any paper previously published on nutrient composition or use of NPMs in Chinese food products.

·      Proper references need to be added every time any law or NPM is mentioned.

·      References are missing in line 297.

·      All across the manuscript, foods exceeding nutrients according to the different NPMs are considered rated/eligible for regulation. This is incorrect since i.e. foods eligible for regulation by the PAHO NPM are those processed and UPF, as stated in the NPM document. However, foods exceeding any of the nutrients are those surpassing the thresholds stablished by the NPM. Please, modify accordingly.

·      Authors should elaborate on the differences among NPMs in previous studies. They should also try to interpret the differences they find.

·      The paragraph on UPF (lines 283-291) is rather poor. Results presented in this work must be put into context with the increase in UPF intake mentioned in the text. Authors have to elaborate on this connection. Just as an example, the health risk may be associated with the nutritional quality of products as shown in this work.

·      Lines 308-309: the high daily salt intake coming from sauces in China is not due to a 4.4-fold greater sodium content compared to sauces in UK, but to the high salt/sodium content. The comparison is not the reason, but the raw content of sodium/salt.

·      Paragraph in lines 311-318: there are many papers published in the last decade on prevalence of NSS. Authors should include some more. I suggest a couple of them in Chile because they present very high rates of NSS:

o  Zancheta, R.C.; Corvalán, C.; Smith, T.L.; Quitral, V.; Reyes, M. Changes in the Use of Non-nutritive Sweeteners in the Chilean Food and Beverage Supply. After the Implementation of the Food Labeling and Advertising Law. Front. Nutr. 2021, 8, 773450.  HYPERLINK "doi:%2010.3389/fnut.2021.773450" doi: 10.3389/fnut.2021.773450

o  Sambra, V.; López-Arana, S.; Cáceres, P.; Abrigo, K.; Collinao, J.; Espinoza A.; Valenzuela, S.; Carvajal, B.; Prado, G.; Peralta, R.; Gotteland, M. Overuse of Non-caloric Sweeteners in Foods and Beverages in Chile: A Threat to Consumers' Free Choice?. Front. Nutr. 2020, 7,  https://doi.org/10.3389/fnut.2020.00068

·      I recommend to include the following reference about the health effect of NSS:

o  World Health Organization. Health effects of the use of non-sugar sweeteners. A systematic review and meta-analysis. 2022. Available online:  HYPERLINK "https://www.who.int/publications/i/item/9789240046429%20" https://www.who.int/publications/i/item/9789240046429 (accessed on 26 May 2022).

·      Lines 335-337: this is an important limitation since the values obtained on the rate of foods with excessive nutrients are misrepresenting the real situation. This is so because some of the foods with no excessive nutrients may be so if all the nutrient information were provided. Therefore, this should be further elaborated in the Discussion and mentioned across all sections of the manuscript (material and methods, results) so that readers may be aware of this at all times.

·      Line 352: Is the value 75% correct?

Author Response

Thank you so much for the very helpful comments. We have carefully considered and responded to each comment and made revisions in the manuscript accordingly. The attachment shows the details for response.

Reviewer 2 Report

In the manuscript titled "Nutritional quality of pre-packaged foods in China under various nutrient profile models" Yuan Li and colleagues, they have reported that various Nutrient Profile Models (NPMs) evaluate the nutritional quality of pre-packaged foods in China to inform future food policy development. Nutrition data for pre-packaged foods were collected through FoodSwitch China from 2017 to 2020. The analyses included 73,885 pre-packaged foods, including 8,236 beverages and 65,649 foods. Processed foods (PFs) and ultra-processed foods (UPFs) accounted for 11.4% and 63.6% of all products, respectively. The overall proportion of products with an excessive quantity of at least one negative nutrient was 67.3% according to the Chilean NPM (2019), 68.6% for the Pan American Health Organization NPM (PAHO NPM), and 81.9% for the Western Pacific Region NPM for protecting children from food marketing (WPHO NPM), respectively. In all NPMs, over half of the food products (55.2%) were identified as containing an excessive quantity of at least one negative nutrient, with higher proportions in UPFs compared to PFs and unprocessed foods. Food groups exceeding nutrient thresholds in most NPMs included snack foods, meat and meat products, bread and bakery products, non-alcoholic beverages, confectionery, and convenience foods. Under all three NPMs, the majority of pre-packaged foods in China would exceed the threshold for at least one negative nutrient, especially for UPFs and PFs. Given the need to prevent obesity and other diet-related chronic diseases, efforts are warranted to improve the healthiness of foods in China through evidence-based food policy. I have a few comments regarding the present manuscript.

-The introduction section looks well designed and has important information.

-Maybe more statistical analyses are required in the section, related to determining patterns or maybe thoroughly analyzing all data.

-What is the importance of the data collected, and what is the main idea to express?

-Maybe a figure thank summarizes the information is required

-Thank you for adding limitations.

Author Response

Thank you for the comments. We have carefully considered and responded to each comment and made revisions in the manuscript accordingly. The attachment shows the details for response.
